# Bi-Level One-Shot Architecture Search for Probabilistic Time Series Forecasting

**Jonas Seng**[1]  **Fabian Kalter**[1]  **Zhongjie Yu**[1]  **Fabrizio Ventola**[1]  **Kristian Kersting**[1, 2, 3, 4]

[1]Technical University of Darmstadt
[2]Centre for Cognitive Science TU Darmstadt
[3]hessian.ai
[4]German Research Centre for AI (DFKI)

**Abstract**  Time series forecasting is ubiquitous in many disciplines. A recent hybrid architecture named predictive Whittle networks (PWNs) tackles this task by employing two distinct modules, a tractable probabilistic model and a neural forecaster, with the former guiding the latter by providing likelihoods about predictions during training. Although PWNs achieve state-of-the-art accuracy, finding the optimal type of probabilistic model and neural forecaster (macro-architecture search) and the architecture of each module (micro-architecture search) of such hybrid models remains difficult and time-consuming. Current one-shot neural architecture search (NAS) methods approach this challenge by focusing on either the micro or the macro aspect, overlooking mutual impact, and could attain the overall optimization only sequentially. To overcome these limitations, we introduce a bi-level one-shot NAS method that optimizes such hybrid architectures simultaneously, leveraging the relationships between the micro and the macro architectural levels. We empirically demonstrate that the hybrid architectures found by our method outperform human-designed and overparameterized ones on various challenging datasets. Furthermore, we unveil insights into underlying connections between architectural choices and temporal features.

## 1 Introduction

Time series forecasting is a predictive task of primary importance that finds applications in several disciplines such as energy management, finance, and healthcare. Recently, next to well-known statistical models like ARIMA, deep neural methods gained popularity due to their representational and predictive power. While achieving good performance across many tasks, most deep neural approaches only provide point estimates without giving insights regarding the uncertainty of their predictions. However, uncertainty about predictions is crucial in many critical applications since it provides additional information that can tell us if the predictions might be erratic. For example, a model predicting that a patient will benefit from a certain treatment should not be trusted if the prediction comes with high uncertainty. Hybrid architectures are a prominent way to obtain high predictive accuracy with useful uncertainty estimations. These architectures generally consist of a *neural forecaster*, *e.g.*, an RNN, and a *probabilistic model*, *e.g.*, isotropic Gaussian distributions (Salinas et al., 2020), normalizing flows (Rasul et al., 2021b), or probabilistic circuits (Yu et al., 2022). In the context of time series, most of these methods operate in the time domain (Salinas et al., 2020; Rasul et al., 2021a,b) while recent works have demonstrated how to leverage the Whittle approximation (Whittle, 1953) to operate in the spectral domain achieving better scaling and taming complexity (Yu et al., 2021a). A recent hybrid architecture named predictive Whittle networks (PWNs) (Yu et al., 2022) exploits predictive uncertainty to further improve forecasting accuracy, outperforming state-of-the-art neural architectures. Thus, PWNs turn out to be a promising venue for probabilistic time series forecasting. However, to date, PWNs have been employed only with

standard neural network architectures, thus, it is unclear whether these hand-crafted architectures are optimally designed for time series forecasting tasks.

Designing a neural architecture from scratch requires often expertise and computational resources. Thus, neural architecture search (NAS) targets the challenge of automatically discovering valuable neural architectures with the goal of substantially reducing the required resources. In the last years, NAS has made significant progress that led to the proliferation of many approaches based, e.g., on genetic algorithms, reinforcement learning, and gradient descent. Despite most NAS algorithms being generally evaluated on well-established tasks such as image classification, recent works have changed the trend and devised NAS approaches tailored to the challenging task of time series prediction (Rakhshani et al., 2020; Chen et al., 2021; Deng et al., 2022). However, existing NAS approaches for time series forecasting only optimize certain parts of an architecture (often referred to as micro- and macro-architecture), therefore, they fail to optimize the *entire* architecture which is crucial to model performance (Santra et al., 2021).

To tackle this shortcoming, we propose a novel differentiable one-shot method that optimizes both the micro- and macro-architectures of PWNs. Our method frames optimization of the micro- and macro-architecture as a bi-level optimization problem and updates the hybrid architecture using gradients from the loss w.r.t. the micro- and macro-architecture in an alternating fashion. Furthermore, we provide an extensive empirical evaluation demonstrating that our bi-level architecture search algorithm significantly improves the prediction accuracy of PWNs by optimizing both its micro- and macro-architecture.[1] To summarize, we make the following contributions:

1. To optimize the entire architecture, we propose a novel bi-level neural architecture search algorithm that optimizes both the macro- and micro-architectures of a hybrid model.

2. We empirically show that optimizing the hybrid PWN architecture using the proposed bi-level architecture search algorithm outperforms hand-crafted baseline architectures to show the effectiveness of our method.

3. To rule out confounding factors, we empirically demonstrate that the performance improvement indeed stems from the optimization, and not from the overparameterization of our models during the search for an optimal architecture.

The rest of this paper is structured as follows. First, we touch upon related work before we formally describe our method. Then, we provide an extensive empirical evaluation before concluding our work.

## 2 Related Work

Our work combines two lines of research, namely probabilistic time series modeling and neural architecture search (NAS). We briefly review both in the following.

**Probabilistic Time Series Modeling.** Conventional time series forecasting models such as long short-term memory (LSTM) (Hochreiter and Schmidhuber, 1997), gated recurrent unit (GRU) (Cho et al., 2014), N-BEATS (Oreshkin et al., 2019), spectral RNN (Wolter et al., 2020), and Informer (Zhou et al., 2021) have shown their great power in computing the prediction $y$ given the context $x$ of a time series. However, these neural forecasters do not naturally provide uncertainty estimates to the predictions, and as a result, users will have difficulty telling how much to trust their predictions.

More recently, based on the development of probabilistic time series models (eg. Gaussian processes (Rasmussen and Williams, 2006; Bruinsma et al., 2020), time series graphical models (Tank et al., 2015), probabilistic circuit based models (Kalra et al., 2018; Trapp et al., 2020; Yu et al., 2021b)), several probabilistic time series forecasting models have been proposed. Neural auto-regressive

---

[1]Code available at `https://github.com/J0nasSeng/Bi-level-optim.git`

models and normalizing flows have been shown to improve predictions (Salinas et al., 2020; Rasul et al., 2021a,b), and modeling the time series in the spectral domain with tractable probabilistic models further improves the forecasting accuracy and provides useful uncertainty estimates(Yu et al., 2022). Although the above-mentioned time series models have shown their predicting power on benchmark datasets, users still need to carefully tune the hyperparameters of the models when given new data sets or new tasks.

**Neural Architecture Search**. As the success of deep learning techniques emerged, the problem of finding optimal architectures for the task at hand became important to solve. Pioneering approaches include the utilization of reinforcement learning (RL) (Pham et al., 2018), evolutionary algorithms (EA) (Real et al., 2017) and the introduction of differentiable neural architectures that allows optimization of architectures using Stochastic Gradient Descent (SGD) (Liu et al., 2019). While RL and EA-based methods find state-of-the-art architectures, they come with high resource consumption, hence often rendering them infeasible, especially for large problems. In contrast, gradient-based approaches are significantly more efficient. Their efficacy relies on approximations of the first-order solution of the bi-level optimization problem these approaches have to solve. Since the major task in evaluating NAS methods on is image classification, recently several works considered NAS specifically in the context of time series data. Deng et al. (2022) designed an AutoML pipeline that optimizes hyperparameters and neural architectures to obtain high-performance time series forecasting models. Chen et al. (2021) and Rakhshani et al. (2020) add time series specific operations in the search space or as pre-processing steps to improve the performance of existing NAS algorithms. The works mentioned above were designed to optimize only one aspect of neural architectures, i.e. the micro- or macro-architecture. Recently, Sun et al. (2022) proposed AGNAS to tackle the joint optimization of micro- and macro-architecture exploiting attention mechanisms to weight operations/blocks of an architecture based on the computed feature representations. This renders AGNAS inapplicable in hybrid architectures that contain components that do not compute feature representations directly (e.g., probabilistic circuits). Thus, we propose a method capable of optimizing both the micro- and macro-architecture jointly, solely requiring a differentiable loss w.r.t. the architecture components rather than relying on feature representations.

## 3 Bi-Level Differentiable Architecture Search

We aim to optimize both, the macro- and micro-architecture of hybrid (neural) models at the same time using a bi-level optimization approach, leading us to a bi-leve architecture search. Therefore, we first briefly describe the hybrid model employed in our paper – predictive Whittle networks (PWNs) – and proceed with our bi-level architecture search algorithm.

### 3.1 Predictive Whittle Networks as Hybrid Models

We focus on the task of time series forecasting with hybrid models such as PWNs, where we are given the context $x$ and need to predict the future $y_{Pred}$ without knowing the ground truth $y_{GT}$. PWNs are a recent hybrid deep learning architectures that model time series in the spectral domain, and aim to both improve prediction accuracy and provide prediction uncertainty estimates. This is done by simultaneously training a neural forecasting network in combination with a Whittle probabilistic circuit (Whittle PC). The Whittle PC allows the tractable computation of inference queries, in our case the Whittle PC computes the log-likelihood of the predictions from the neural forecaster given the forecaster's context. This information is then used as feedback to the neural forecaster, to further improve the prediction accuracy. This is achieved by using the Whittle forecasting loss ($\mathcal{L}_{\mathrm{WF}}$), which can be described as a prediction error ($\mathcal{L}_{\mathrm{Pred}}$, *e.g.* mean squared error) (re-)weighted by the Whittle likelihood ($\mathcal{L}_{\mathrm{WLL}}$)

$$\mathcal{L}_{\mathrm{WF}} = \mathcal{L}_{\mathrm{Pred}}(y_{Pred}, y_{GT}) \cdot \mathcal{L}_{\mathrm{WLL}}(x, y_{Pred}, y_{GT}). \tag{1}$$

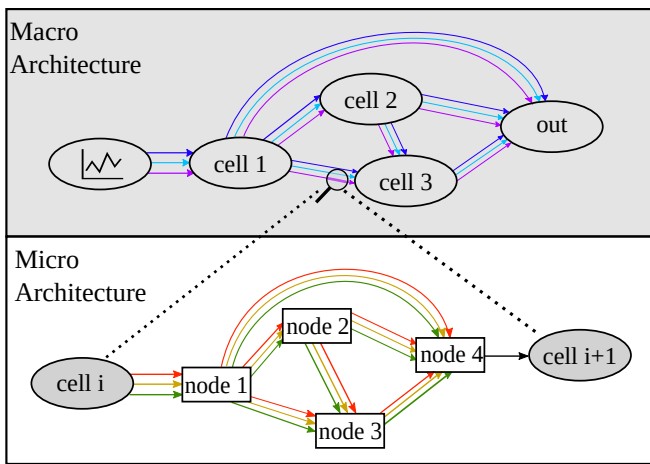

Figure 1: **Bi-Level Architecture Search Space**. Our method optimizes both the macro- and micro-architectures simultaneously. The search space is thus defined as a bi-level search space where each edge of the macro search space (top) is a neural network whose architecture – called micro-architecture – is optimized as well (bottom). Ellipses denote representations by operations in the macro-architecture (*i.e.* neural networks) while rectangles denote representations computed by operations in the micro-architecture.

While the prediction error $\mathcal{L}_{\text{Pred}}$, measures the deviation between the prediction and the ground truth, the Whittle likelihood indicates how likely the prediction fits its distribution. The Whittle likelihood models time series in the spectral domain. More specifically, the Fourier coefficients of a time series are assumed to follow a multivariate Gaussian distribution, and modeled with a Whittle PC for computing the Whittle likelihood. We refer to Yu et al. (2021a, 2022) for a detailed discussion. Furthermore, in order to accelerate the training process and extract the valuable likelihood feedback efficiently, a warm-up phase is utilized, resulting in the following loss function

$$\mathcal{L} = (1 - \beta)\mathcal{L}_{\text{Pred}}(y_{Pred}, y_{GT}) + \beta\,\mathcal{L}_{\text{WF}}(x, y_{Pred}, y_{GT}), \qquad (2)$$

where the parameter $\beta$ is linearly increased during training. We refer to Yu et al. (2022) for more details of the PWN model.

The architecture of the PWN in Yu et al. (2022) is hand-crafted. In the following we show how to automatically search for 1) the combination of the neural components (the macro-architecture) and 2) the architecture of each given neural component itself (the micro-architecture).

## 3.2  Problem Statement and Search Procedure

Inspired by Liu et al. (2019) we view both, the micro- and macro-architecture as directed acyclic graphs (DAGs) $\mathcal{G}_{\text{macro}}$ and $\mathcal{G}_{\text{micro}}$ respectively, each containing any number of nodes in addition to an input and an output node. In both DAGs, each node is connected to all of its predecessors. The edges represent possible operations from a set of operations $\mathcal{O}_{\text{macro}}$ and $\mathcal{O}_{\text{micro}}$ respectively. In the case of the micro-architecture, operations usually represent small building blocks such as convolutions while in the case of the macro-architecture, operations represent cells from the micro-level. The general overview of our algorithm is visualized in Figure 1. In the case of PWN, the macro-architecture includes the PWN building blocks, *i.e.* different neural forecasters and different choices of PCs.

**Macro-Architecture Search.** The set of possible macro-architectures of PWNs is represented as a DAG $\mathcal{G}_{\text{macro}}$. The set of operations $\mathcal{O}_{\text{macro}}$ consists of modules parameterized as neural networks.

The input of a cell is defined as the weighted sum of the outputs of its predecessors, with an macro-architecture weight $\omega_m$ for an edge with module $m \in \mathcal{O}_{\text{macro}}$

$$\bar{m}(x) = \sum_{m \in \mathcal{O}_{\text{macro}}} \frac{\exp(\omega_m)}{\sum_{m' \in \mathcal{O}_{\text{macro}}} \exp(\omega_{m'})} m(x). \tag{3}$$

The architecture weights indicate which module is expected to best optimize the loss w.r.t. the macro-architecture. We relax the categorical search to a continuous space by computing the softmax over all possible modules. The objective of PWN in eq. (2) then becomes

$$\mathcal{L} = (1 - \beta)\mathcal{L}_{\text{Pred}}(\bar{m}_{Pred}(x), y_{GT}) + \beta \mathcal{L}_{\text{WF}}(x, \bar{m}_{Pred}(x), y_{GT}), \tag{4}$$

where $\bar{m}_{Pred}(x)$ represents the weighted predictions from the neural forecaster building blocks, $x$ is the original input (context) in the time domain, and $y_{GT}$ is the ground truth prediction in the time domain. We define the objective of the macro-architecture search for modules parameterized by $\omega$ as

$$\min_{\omega} \mathcal{L}_{val}(w^*(\omega), \omega), \tag{5}$$

where $w^*(\omega) = \arg\min_w \mathcal{L}_{train}(w, \omega)$. This two-step optimization loop is computationally expensive and inefficient as one single step for the macro-architecture weights requires a full optimization of the entire network. To tackle this, we follow Liu et al. (2019) and approximate the optimization task with one-step gradient descent.

$$\begin{aligned} &\nabla_{\omega}\mathcal{L}_{val}(w^*(\omega), \omega) \\ &\approx \nabla_{\omega}\mathcal{L}_{val}(w - \xi\nabla_w\mathcal{L}_{train}(w, \omega), \omega). \end{aligned} \tag{6}$$

In this way, we are able to efficiently obtain the macro-architecture weights that indicate which modules to use for the PWN model.

**Micro-Architecture Search**. After discussing the macro-architecture search, we now complement our method by the optimization of the micro-architecture, leading to our bi-level architecture search. Similar to the macro-architecture we represent the set of possible micro-architectures a DAG $\mathcal{G}_{\text{micro}}$. Each module $m \in \mathcal{O}_{\text{macro}}$ has its own associated DAG $\mathcal{G}^m_{\text{micro}}$, inducing a hierarchy of graphs. Let us denote the architecture weights of the neural spectral forecaster and Whittle PC as $\omega_N$ and $\omega_P$ respectively. For better understanding and readability the level of a corresponding weight is indicated by 1 for the macro-level and 2 for the micro-level.

We can now rewrite Eq. 4 in terms of the micro-architectures of all modules since these fully define the architecture of each module

$$\mathcal{L} = (1 - \beta)\mathcal{L}_{\text{Pred}}(\omega^1_N, y_{GT}) + \beta \mathcal{L}_{\text{WF}}(\omega^1_N, \omega^1_P) \tag{7}$$

to a bi-level view

$$\mathcal{L} = (1 - \beta)\mathcal{L}_{\text{Pred}}(\omega^1_N, \omega^2_N) + \beta \mathcal{L}_{\text{WF}}(\omega^1_N, \omega^2_N, \omega^1_P, \omega^2_P). \tag{8}$$

The spectral forecaster weights affect both parts of the loss as the time series prediction is part of calculating the Whittle likelihood, while the Whittle PC only influences said likelihood. We can now also extend the formulation of our optimization objective to the bi-level problem and optimize both levels in union

$$\min_{\omega^1, \omega^2} \mathcal{L}_{val}(w^*(\omega^1, \omega^2), \omega^1, \omega^2), \tag{9}$$

As in the macro search space, each node in $\mathcal{G}_{\text{micro}}$ is defined as a weighted sum over operations:

$$\bar{m}(x) = \sum_{m \in \mathcal{O}_{\text{micro}}} \frac{\exp(\omega^2_m)}{\sum_{m' \in \mathcal{O}_{\text{micro}}} \exp(\omega^2_{m'})} m(x). \tag{10}$$

With that, we can calculate the final output of $\mathcal{G}_{\text{micro}}$ by feeding the weighted operations $\bar{m}(x)$ as input into all following nodes of $\mathcal{G}_{\text{micro}}$. The final output is then calculated as the mean of all outputs $\bar{m}_{micro}(x) = \frac{1}{n} \sum_{i=1}^{n} \bar{m}_i(x)$, and can be used to replace a corresponding $m(x)$ in equation 3. This allows us to include the set of micro-architecture weights in our optimization of the macro-architecture weights to enable our algorithm to update both levels in union.

We now can state our optimization strategy to optimize the architecture on both, the macro- and micro-level as stated in Eq. 9. As all aspects of the architecture are relaxed to a continuous space, gradients can be computed w.r.t. the macro- and micro-architecture. The model- and architecture parameters are updated in an alternating fashion. Model parameters are updated using standard SGD with the architecture parameters kept fixed. To update the architecture, we extend Eq. 6:

$$
\begin{aligned}
&\nabla_{\omega^1, \omega^2} \mathcal{L}_{val}(w^*(\omega^1, \omega^2), \omega^1, \omega^2) \\
&\approx \nabla_{\omega^1, \omega^2} \mathcal{L}_{val}(w - \xi \nabla_w \mathcal{L}_{train}(w, \omega^1, \omega^2), \omega^1, \omega^2)
\end{aligned}
\tag{11}
$$

The architecture parameters are updated with SGD on validation data to foster the discovery of well-generalizing architectures.

## 4 Experiments

We now examine how the differentiable bi-level architecture search performs on various time series forecasting datasets. The following research questions are tackled:

**(Q1)** How do PWN architectures found by our algorithm perform compared to manually designed architectures?

**(Q2)** How reliably does our algorithm choose the optimal PWN macro-architecture across datasets?

**(Q3)** How do different datasets influence the optimized micro-architectures?

### 4.1 Experimental Setup

**Datasets**. To evaluate the effectiveness of the proposed algorithm, we use 5 real-world time series data sets for empirical experiments. The well-known `M4` competition dataset (Makridakis et al., 2020) consists of multiple time series from different domains such as finance and demographics. Moreover, the time series in `M4` cover a large range of sampling frequencies, from hourly to yearly, making it a challenging dataset for time series forecasting. The `Power` dataset contains power consumption data from different EU power grids sampled with a frequency of 15 minutes, from 2011 to 2019 (Wolter et al., 2020). The `Exchange` dataset is a collection of daily exchange rates from 8 countries from 1990 to 2016 (Lai et al., 2018). The `Wiki` dataset reports the daily hits of 2000 Wikipedia pages (Gasthaus et al., 2019; Rasul et al., 2021b). The `Solar` dataset, depicts the solar energy production of 137 plants in the state of Alabama in 2006 (Lai et al., 2018). A comprehensive overview over different properties of the data sets can be found in Appendix A.

**Search Space**. The search space in NAS tasks is crucial and thus needs to be carefully designed. In our experiments, the macro-architecture search space consists of the PWN building blocks: SRNN, STrans, CWSPN, and WEin. The micro-architecture search space for the SRNN component consists of 8 cells and 5 operations, where each cell is a GRU cell implemented in Yu et al. (2022). The operations include a linear transformation followed by *sigmoid*, *tanh* or *ReLU* activation functions, as well as *none* and *identity* connections. The CWSPN search space, on the other hand, is designed with 4 nodes, as this network tends to be smaller than the SRNN. The 6 operations contain convolutions and dilated convolutions with kernel sizes 3 and 5, as well as *identity* and *none* connections. Table 1 provides a general overview of the search space in our experiments. We set the architecture learning rate to 0.0003 and the weight decay to 0.001 in our experiments.

| Component | Level | Options | #Options |
|---|---|---|---|
| Model Structure | Macro | [SRNN, STrans] | 2 |
| | | [CWSPN, WEin] | 2 |
| SRNN | Micro | [*ReLU, sigmoid, tanh, none, identity*] | 5 |
| CWSPN | Micro | [3x1 (dil) *conv*, 5x1 (dil) *conv*, *none*, *identity*] | 6 |

Table 1: **Search Space Overview.** We define our search space across a macro-architectural and a micro-architectural level. Optimizing the macro-architecture can be considered as selecting an appropriate search space while optimization of the micro-architecture performs the actual optimization of neural architectures.

| | SMAPE % | | | | | | MSE | | | |
|---|---|---|---|---|---|---|---|---|---|---|
| | M4 Yearly | M4 Quarterly | M4 Monthly | M4 Weekly | M4 Daily | M4 Hourly | Power $10^5$ | Exchange $10^{-4}$ | Wiki $10^7$ | Solar $10^3$ |
| TFT | 15.03 ± 0.12 | 17.22 ± 0.05 | 16.55 ± 0.03 | 16.65 ± 0.08 | 16.68 ± 0.09 | 16.99 ± 0.04 | 34.07 ± 0.10 | 14.72 ± 0.13 | 11.16 ± 0.05 | 229 ± 71.01 |
| ETS | 15.10 ± 0.00 | 17.46 ± 0.00 | 19.46 ± 0.00 | 19.93 ± 0.00 | 19.39 ± 0.00 | 18.65 ± 0.00 | 34.54 ± 0.00 | 10.89 ± 0.00 | 10.90 ± 0.00 | 1156 ± 0.00 |
| DeepAR | *12.31* ± 0.08 | 11.85 ± 0.06 | *10.71* ± 0.09 | 10.20 ± 0.04 | 10.52 ± 0.02 | 10.93 ± 0.04 | 37.21 ± 0.06 | 12.88 ± 0.04 | 7.44 ± 0.07 | 219 ± 43.06 |
| SRNN & CWSPN | 14.60 ± 0.11 | 11.43 ± 0.09 | 14.71 ± 0.07 | 4.76 ± 0.10 | 6.16 ± 0.06 | 5.44 ± 0.12 | 4.14 ± 0.16 | 2.08 ± 0.28 | 5.38 ± 0.17 | 1.61 ± 0.34 |
| SRNN & WEin | 14.51 ± 0.14 | 11.31 ± 0.08 | 14.96 ± 0.11 | 4.75 ± 0.09 | 6.34 ± 0.10 | 5.48 ± 0.09 | 4.37 ± 0.08 | *1.89* ± 0.27 | 5.23 ± 0.17 | 1.58 ± 0.29 |
| STrans & CWSPN | 15.51 ± 0.16 | 12.06 ± 0.11 | 14.94 ± 0.11 | 4.67 ± 0.10 | 6.55 ± 0.10 | 5.37 ± 0.08 | 4.11 ± 0.19 | 6.31 ± 1.21 | 5.28 ± 0.10 | 1.51 ± 0.21 |
| STrans & WEin | 14.92 ± 0.15 | 11.36 ± 0.08 | 14.47 ± 0.13 | 4.58 ± 0.09 | 6.06 ± 0.09 | 5.41 ± 0.09 | 4.05 ± 0.14 | 6.32 ± 0.99 | 5.38 ± 0.28 | 1.59 ± 0.25 |
| SRNN$^+$ & CWSPN | 14.86 ± 0.16 | 11.82 ± 0.18 | 15.02 ± 0.20 | 5.02 ± 0.19 | 7.86 ± 0.30 | 6.56 ± 0.28 | 6.89 ± 0.30 | 233 ± 16.27 | 5.09 ± 0.33 | 2.36 ± 0.38 |
| PWN$^\star$ (ours) | **14.11** ± 0.36 | *11.11* ± 0.24 | **14.09** ± 0.32 | *4.53* ± 0.25 | *5.92* ± 0.40 | *5.08* ± 0.31 | *3.82* ± 0.16 | 1.95 ± 0.72 | *4.80* ± 0.41 | *1.34* ± 0.49 |

Table 2: **Accuracy Results**. We compare the architecture found by our bi-level architecture search with all combinations of macro-architectures, neural baselines (TFT, DeepAR), and a statistical baseline model (ETS). Lower values are better. The best result with uncertainty-guided training is marked in **bold**, and the overall best result is in *italic*. Our algorithm (marked with $\star$) outperforms uncertainty-guided hybrid baseline models on all except one dataset. The selection of the correct macro-architecture critically influences the performance of the final model, rendering optimization of the macro-architecture crucial. For more details regarding model training please refer to appendix B.

## 4.2 (Q1) Time Series Forecasting Results

We now are ready to quantitatively compare the time series forecasting results with the baselines. For our evaluation, we first applied our bi-level architecture search to obtain an optimized PWN architecture denoted by PWN$^\star$. Then, the found architecture was trained from scratch and evaluated on the test set against the baseline models. As baseline models, we trained all possible combinations of the macro-search space and fixed the micro-architecture to the same architecture used in (Yu et al., 2022). Additionally, Exponential Smoothing (ETS) (Gardner, 2006), Deep AR (Salinas et al., 2020) and Temporal Fusion Transformer (TFT) (Lim et al., 2020) served as baselines. The time series forecasting results from PWN$^\star$ together with our baselines – the 4 handcrafted PWN structures – are depicted in table 2. On each dataset, we repeated the experiment 10 times and report the average and the standard deviation.

We find that the architecture PWN$^\star$ found by our bi-level architecture search outperforms the baselines that are guided by uncertainty during training on M4, Power, Wiki and Solar datasets. Also, PWN$^\star$ is very competitive on the Exchange dataset as runner-up. Further, we observe a slight increase in standard deviation for PWN$^\star$, likely originating from random effects during architecture search. Our results indicate that a bi-level architecture search algorithm can indeed improve the performance of hybrid models, by finding the best modules as well as the suitable structures of each module.

In order to evaluate the influence of the increased number of parameters in PWN$^\star$ during the search, we conduct another experiment by employing a larger SRNN s.t. the number of parameters is close to the model used during the search. The larger model has an SRNN with 12 GRU cells, denoted as SRNN$^+$ & CWSPN, shown in the second last row in table 2. The results confirm that, generally, simply enlarging the model size does not yield an improvement in performance, as all the prediction results from SRNN$^+$ & CWSPN are worse than our bi-level searched PWN$^\star$. Therefore, we can conclude that the improvement of PWN$^\star$ is indeed from architecture search, rather than by only increasing the model size.

To conclude, our results show that we outperform the handcrafted models with our bi-level optimization algorithm on all but one data set. Additionally, we have shown that this increase is not solely based on larger models but the result of a more optimal architecture.

### 4.3 (Q2) Optimizing the Macro-Architecture

We now evaluate the module selection performance of the search over the macro-architecture search space. The results from all combinations of modules given the original PWN are shown in table 2, and the best modules are summarised in table 3 labeled 'best'. The proposed macro-architecture search algorithm successfully selects the module of the spectral neural forecaster, as they mostly match the best modules. However, the macro-architecture search could not find the best module of the Whittle PC in most of the cases. One possible explanation is that due to their own structural properties, CWSPN converges faster during training than WEin using gradient descent. It is well known that gradient-based architecture search tends to select architectures with higher convergence speed rather than selecting architectures minimizing the objective. This could lead the macro-architecture search to select faster converging modules.

To conclude, our method selects the best neural forecasting module in most cases, the selection of the best-performing PC remains challenging and is one potential future research direction. Additionally, table 2 clearly shows that choosing the correct macro-architecture is crucial to obtaining high-performing models. Our approach consistently identifies the best neural forecaster module, allowing it to find state-of-the-art model architectures.

|  |  | M4 Yearly | M4 Quarterly | M4 Monthly | M4 Weekly | M4 Daily | M4 Hourly | Power | Exchange | Wiki | Solar |
|---|---|---|---|---|---|---|---|---|---|---|---|
| spectral | chosen | STrans | STrans | STrans | STrans | STrans | STrans | SRNN | SRNN | STrans | SRNN |
| forecaster | best | SRNN | SRNN | STrans | STrans | STrans | STrans | SRNN | SRNN | STrans | SRNN |
| Whittle | chosen | CWSPN | CWSPN | CWSPN | CWSPN | CWSPN | CWSPN | CWSPN | CWSPN | CWSPN | CWSPN |
| PC | best | WEin | WEin | WEin | WEin | WEin | WEin | CWSPN | WEin | CWSPN | WEin |

Table 3: **Module Selection**. Comparison of module search choice to the best-performing combination. Identical color means, that the correct module has been selected, according to our baseline results. We can see that the selection of the spectral forecaster works well, while the performance of PC selection is lacking.

### 4.4 (Q3) Optimizing the Micro-Architecture

After investigating the performance of our bi-level architecture search to find the best macro-architecture, we proceed by analyzing its behavior in terms of finding micro-architectures. We analyze the frequency at which each operation is selected across different datasets. This allows us to draw conclusions about how sensitive the architecture selection of our algorithm is w.r.t. the data presented. For our analysis, we ran the bi-level architecture search for 5 runs with different random seeds, and illustrate the average number of operations for SRNN (in table 4) and CWSPN (in table 5). We refer to Appendix G for better visualization of the tables.

We analyze the micro-architectures of the SRNN and CWSPN separately. The micro-architecture search results on SRNN, shown in table 4, demonstrate that the preferred operations differ between data sets. First of all, *identity* connections are chosen vastly more than other operations, on

|          | M4 Yearly | M4 Quarterly | M4 Monthly | M4 Weekly | M4 Daily | M4 Hourly | Power | Exchange | Wiki | Solar |
|----------|-----------|--------------|------------|-----------|----------|-----------|-------|----------|------|-------|
| none     | 0         | 0.2          | 0.4        | 0.4       | 0        | 0         | 1.2   | 0        | 0.2  | 0.2   |
| tanh     | 0         | 0            | 0          | 0         | 0        | 0         | 0.2   | 0        | 0    | 0     |
| *ReLU*   | 0.8       | 0.4          | 4.4        | 6.4       | 12.2     | 11.0      | 5.6   | 0        | 0    | 16.6  |
| sigmoid  | 6.4       | 7.6          | 1.0        | 0.6       | 0.6      | 2.2       | 0     | 1.4      | 4.6  | 0     |
| identity | 38.6      | 27.8         | 30.2       | 28.6      | 23.2     | 22.8      | 27.0  | 30.6     | 31.2 | 17.2  |

Table 4: **SRNN architecture results**. Average number of times an operation is present in the optimized SRNN architecture. We can observe a strong preference for identity connections and a tendency of either *sigmoid* or *ReLU* activation functions.

most of the data sets. This behavior coincides with the behavior of DARTS and has already been discussed in Heuillet et al. (2023). *Identity* connections fundamentally represent residual connections, providing an initial performance boost. This initial advantage is often enough to choose this operation over others, even though its long-term performance might be subpar. On the other hand, *none* connections have been chosen very rarely on all the data sets, which results in much larger model sizes compared to the standard SRNN structure. Secondly, when looking at the activation function operations, one can see that *ReLU* and *sigmoid* are the most selected operations on all the data sets. Interestingly, we observe that *ReLU* operations are chosen more frequently on subsets of the M4 data set with higher sampling rates (weekly, daily and hourly), as well as on Power (15min) and Solar (1h). Similarly, *sigmoid* operations are more preferred by the data sets with a relatively lower sampling rate, such as Exchange and Wiki (1 day) and M4 (yearly and quarterly). We leave further investigation of this behavior for future work.

|             | M4 Yearly | M4 Quaterly | M4 Monthly | M4 Weekly | M4 Daily | M4 Hourly | Power | Exchange | Wiki | Solar |
|-------------|-----------|-------------|------------|-----------|----------|-----------|-------|----------|------|-------|
| none        | 3.4       | 2.8         | 3.6        | 3.0       | 2.4      | 2.4       | 3.0   | 0.4      | 0.8  | 0     |
| identity    | 0         | 0           | 0          | 0         | 0        | 0         | 0.4   | 8.0      | 0    | 3.6   |
| 3x1 conv    | 0.2       | 0           | 4.0        | 3.0       | 3.4      | 3.6       | 2.4   | 0.4      | 4.8  | 3.8   |
| 5x1 conv    | 0         | 0.2         | 1.0        | 2.4       | 0.4      | 0.2       | 1.6   | 0.2      | 1.4  | 0     |
| 3x1 dil conv| 3.2       | 3.8         | 0          | 0.8       | 0.8      | 0.2       | 0     | 0        | 0.6  | 0     |
| 5x1 dil conv| 3.2       | 3.2         | 1.0        | 2.0       | 3.0      | 3.0       | 3.2   | 0        | 2.4  | 3.8   |

Table 5: **CWSPN architecture results**. Average number of times an operation is present in the optimized CWSPN architecture. We can observe that data sets, with the exception of Exchange, either prefer a combination of 3x1 dilated and 5x1 dilated convolutions or 3x1 and 5x1 dilated convolutions.

Similarly, the micro-architecture search results on CWSPN are shown in table 5. The most notable observation is that the convolution operations are much less frequently selected on the smallest data set Exchange, which might indicate that for data sets with smaller sizes, the fully connected layers are already sufficient. Furthermore, the zero operations are likely to be selected on M4 and Power datasets, whose size are both relatively larger than the rest, resulting in smaller model sizes. Lastly, we also observe that the sampling rate and the length of the context time series also influence the selection of operations. More specifically, the M4 subsets Yearly and Quarterly, which both have a low sampling rate and short context length, tend to select more operations of 3× and 5× dilated convolution, while the rest subsets of the M4 data set are given more operations of 3× and 5× convolution.

To summarize, we find that the data presented has a higher impact on the CWSPN micro-architectures than on the SRNN micro-architectures since in the case of SRNN the operation count is quite similar across datasets. This is not the case for CWPSN architectures. Further, we found that our method seems to favor identity connections during the search, similar to the findings in (Heuillet et al., 2023). Finally, we find that the sampling frequency has a notable effect on the choice of convolution operations in CWSPNs.

## 5 Conclusion

In this work, we proposed a novel differentiable neural architecture search capable of optimizing both, the micro- and macro-architecture of hybrid models. We evaluated our algorithm across various time series forecasting datasets using a search space derived from a recent state-of-the-art hybrid model (PWNs) for probabilistic time series forecasting. Empirical results show that our bi-level architecture search algorithm finds suitable micro- and macro-architectures, and in turn, outperforms our baselines on time series forecasting.

**Limitations & Future Work**. One of the limitations of this work is that we did not prove that our bi-level architecture search favors operations with faster convergence properties over operations minimizing the objective function. Probing this conjecture is therefore a good direction for future work as well as exploring how to balance the module selection with their convergence in training if our conjecture holds true. Also, besides inheriting the efficiency of DARTS, we also inherit possible unstable behaviors in some cases. An analysis of the sensitivity of our approach w.r.t. the choice of hyperparameters is an interesting venue for future work as well as exploring other, more robust extensions of DARTS in our framework. Another natural extension is to enable micro-architecture search for other modules, *e.g.* STrans and WEin, as well as to explore other modules that are even not evaluated in the original PWN.

## 6 Broader Impact Statement

After careful reflection, the authors have determined that this work presents no notable negative impacts to society or the environment.

**Acknowledgements**. This work was supported by the National High-Performance Computing Project for Computational Engineering Sciences (NHR4CES) and the Federal Ministry of Education and Research (BMBF) Competence Center for AI and Labour ("KompAKI", FKZ 02L19C150). Furthermore, this work benefited from the cluster project "The Third Wave of AI".

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

## A  Dataset Properties

In this section, we provide the detailed statistics of the datasets used in our experiments.

|  | Dimension | Domain | Freq. | Time Steps per Dimension | Prediction Length | Context Timespan | Evaluation Metric |
|---|---|---|---|---|---|---|---|
| M4 Yearly | 23000 | $\mathbb{R}^+$ | 1-y | 30 | 6 | 24 | SMAPE |
| M4 Quarterly | 24000 | $\mathbb{R}^+$ | 3-mon | 40 | 8 | 32 | SMAPE |
| M4 Monthly | 48000 | $\mathbb{R}^+$ | 1-mon | 469* | 18 | 108 | SMAPE |
| M4 Weekly | 359 | $\mathbb{R}^+$ | 1-wek | 1455* | 14 | 63 | SMAPE |
| M4 Daily | 4227 | $\mathbb{R}^+$ | 1-d | 1006* | 14 | 70 | SMAPE |
| M4 Hourly | 414 | $\mathbb{R}^+$ | 1-hr | 960* | 48 | 480 | SMAPE |
| Power | 8 | $\mathbb{R}^+$ | 15-min | 139872 | 144 | 1440 | MSE |
| Exchange | 8 | $\mathbb{R}^+$ | 1-d | 6071 | 30 | 180 | MSE |
| Wiki | 2000 | $\mathbb{N}_0$ | 1-d | 792 | 30 | 180 | MSE |
| Solar | 137 | $\mathbb{R}^+$ | 1-hr | 7009 | 24 | 720 | MSE |

Table 6: **Dataset Parameters**. Overview of dataset properties and fundamental Hyperparameters.
\* Dimensions have variable length, highest number of time steps per dimension is reported.

## B  Hyperparameters

### B.1  PWN

In this section we provide the hyperparameters for PWN training. All experiments were run with seeds 0-10.

|  | M4 Yearly | M4 Quarterly | M4 Monthly | M4 Weekly | M4 Daily | M4 Hourly | Power | Exchange | Wiki | Solar |
|---|---|---|---|---|---|---|---|---|---|---|
| #Epochs | 15000 | 15000 | 15000 | 15000 | 15000 | 15000 | 5000 | 1000 | 1000 | 200 |
| Batchsize | 256 | 256 | 256 | 256 | 256 | 64 | 256 | 32 | 64 | 64 |
| FFT Window Size | 6 | 8 | 18 | 14 | 14 | 24 | 96 | 60 | 60 | 24 |
| FFT Compression | 1 | 1 | 1 | 1 | 1 | 1 | 4 | 2 | 2 | 2 |
| Hidden dim. | 128 | 196 | 196 | 64 | 196 | 64 | 64 | 64 | 64 | 64 |
| Layers | 2 | 3 | 3 | 2 | 2 | 2 | 2 | 2 | 2 | 2 |
| EM step-size | 0.05 | 0.025 | 0.05 | 0.05 | 0.05 | 0.05 | 0.05 | 0.05 | 0.05 | 0.05 |
| EM freq. | 1 | 1 | 5 | 1 | 5 | 1 | 1 | 1 | 1 | 1 |
| Learning Rate | 0.001 | 0.001 | 0.001 | 0.001 | 0.001 | 0.001 | 0.004 | 0.003 | 0.004 | 0.003 |

Table 7: Overview of PWN hyperparameters.

### B.2  Baselines

The baselines were trained using the AutoGluonTS and neuralforecast library. All baselines optimized the SMAPE loss for M4 and the MSE loss for the other datasets. The prediction length was set according to Tab. 6. We applied a basic hyperparameter optimization for each model provided by AutoGluonTS/neuralforecast.

## C  Computational Resources and Footprint

Our bi-level approach makes use of the differentiability of the loss w.r.t. the architecture on both the micro- and macro level. The architecture optimization follows Liu et al. (2019), thus, it exploits weight sharing and a one-step approximation of the model weights. As DARTS is an efficient

optimization method, and since we apply it on two levels of the architecture, our approach inherits the time requirements of DARTS.

Most experiments were conducted on machines with an Nvidia RTX 2080 Ti GPU and an AMD Ryzen 7 3800X CPU. Ablation experiments have been conducted on Nvidia DGX machines with A100 GPUs. The total compute time for all experiments (including baselines) is approximately 5000 GPU hours.

On A100 GPUs, the runtime of one search is approximately 7 hours on average while the training of the found PWN architecture takes about 2.5 GPU days. The maximum memory usage during the experiments was 5GB of RAM (on GPU).

In an additional ablation study we only search the macro- and then the micro-architecture. In this setting, the total running time to perform the search on both levels is about 13-15 hours (6.5–7.5 hours for each level), almost double the time required by our bi-level search. Therefore, compared to sequentially first searching for a macro-architecture and then for a micro-architecture (given the selected macro-architecture), by searching both simultaneously with our bi-level approach, we nearly halved the search time. The main computational load is still the full training of the found PWN architecture, which needs 2.5 GPU days.

## D  SMAPE

The symmetric mean absolute percentage error (SMAPE) can be calculated as

$$SMAPE = \frac{100}{n} \sum_i \frac{|\hat{y}_i - y_i|}{|\hat{y}_i| + |y_i|}, \tag{12}$$

where $\hat{y}$ are the predicted values and $y$ refers to the ground truth.

## E PWN Module Search Schematics

In this section, we demonstrate the graph for bi-level architecture search space for PWN.

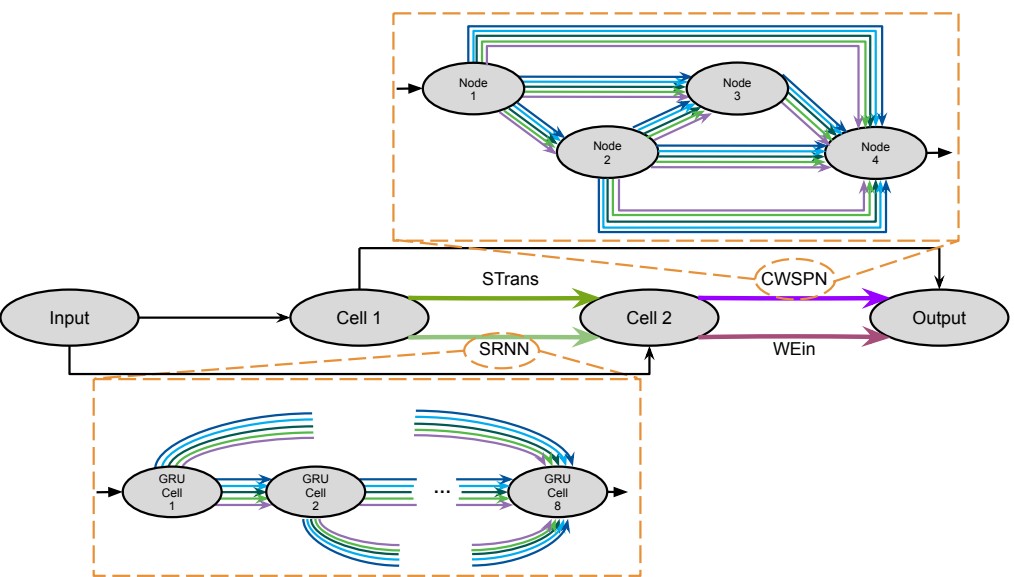

Figure 2: **Bi-Level PWN Architecture Search Space**. Depicted is the overview of the architecture search space for PWN.

## F Bi-Level Optimization Algorithm

**Algorithm 2 Bi-Level Architecture Search.** We alternate between updating the model- and architecture weights during optimization (steps 2 and 4). In steps 1 and 2, model weights $w$ are updated with a fixed architecture. Therefore, we compute the output of the network by weighting the output of the neural forecasters and Whittle PCs according to the current macro-architecture weights $\sigma(\omega^1)$ where $\sigma$ is the softmax function. Each forecaster network $N$ itself is a supernet (representing the micro-architecture via weights $\omega^2$), here the same mechanism is applied. After that the training loss is computed and model weights $w$ are updated. In steps 3 and 4 the same procedure is applied. However, the validation loss is computed and architecture weights are updated.

---

**Input** $x$, weight decay $\rho$, architectural learning rate $\gamma$, ground truth $y$

1. Pass input $x$ from training set through macro- and micro-level
$out_N = \sum_{N\in\text{forecasters}} \sigma(\omega^1)_N \cdot N(x, \omega^2_N, w_N)$;
$out_P = \sum_{P\in\text{Whittle PCs}} \sigma(\omega^1)_P \cdot P(x, out_N, w_P)$;
$\mathcal{L}_{train} \leftarrow \mathcal{L}_{WF}(x, out_N, out_P, y)$

2. Update network weights of all micro-level networks with
$w = w - \xi\nabla_w\mathcal{L}_{train}(w, \omega^1, \omega^2) + \rho||w||$ where $w = \bigcup_{N\in\text{forecasters}} w_N \cup \bigcup_{P\in\text{Whittle PCs}} w_P$

3. Pass input $x$ from validation set through macro- and micro-level
$out_N = \sum_{N\in\text{forecasters}} \sigma(\omega^1)_N \cdot N(x, \omega^2_N, w_N)$;
$out_P = \sum_{P\in\text{Whittle PCs}} \sigma(\omega^1)_P \cdot P(x, out_N, w_P)$;
$\mathcal{L}_{val} \leftarrow \mathcal{L}_{WF}(x, out_N, out_P, y)$

4. Update micro- and macro-level architecture weights of all micro-level networks with
$(w^1, w^2) = (w^1, w^2) - \gamma\nabla_{\omega^1,\omega^2}\mathcal{L}_{val}(w - \xi\nabla_w\mathcal{L}_{train}(w, \omega^1, \omega^2), \omega^1, \omega^2) + \rho||(\omega^1, \omega^2)||$
where $w$ is defined as in step 2.

---

# G  Architecture Composition

The Example Appendix can be removed.

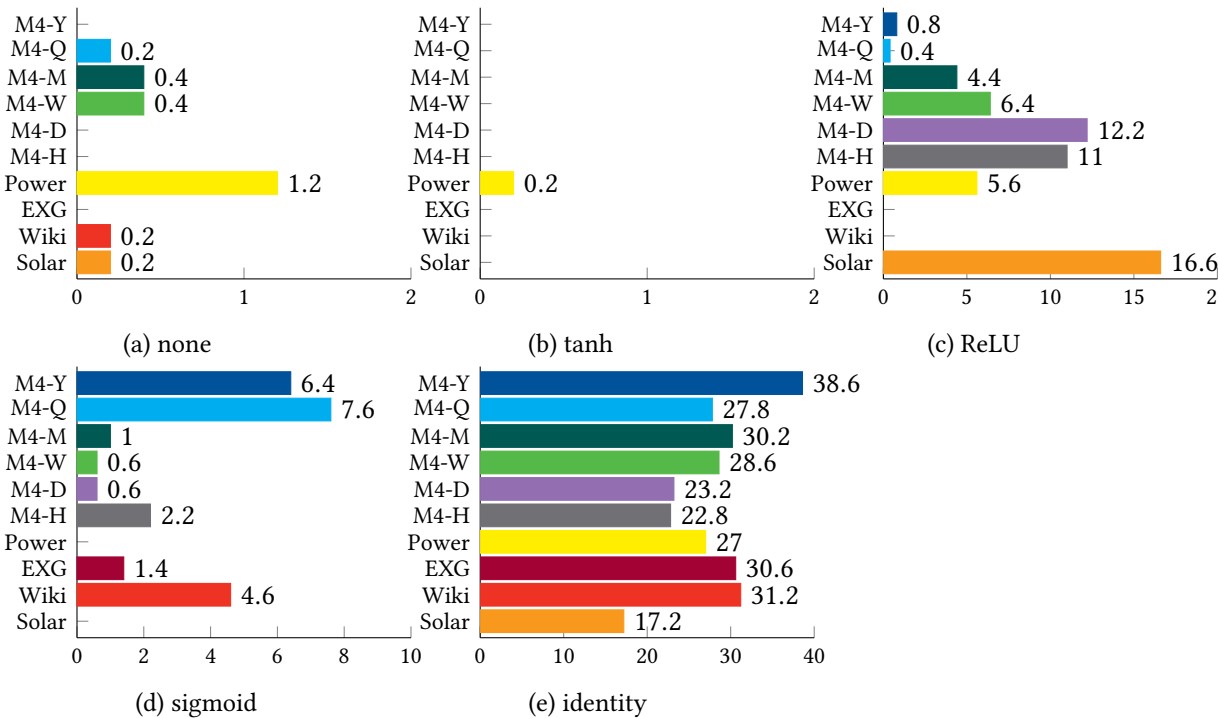

Figure 3: SRNN Architecture composition bar plot.

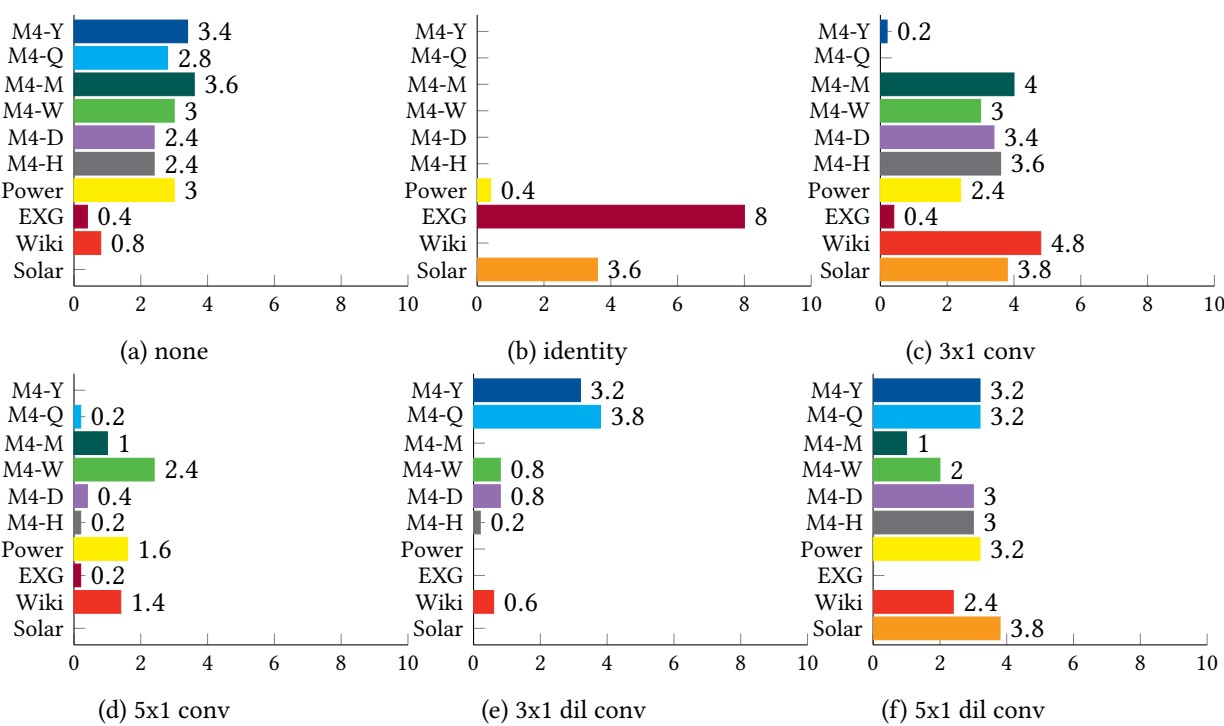

Figure 4: CWSPN Architecture composition bar plot.

