# OpenReview forum: "Bi-Level One-Shot Architecture Search for Probabilistic Time Series Forecasting"
_automl.cc/AutoML/2024/Conference — AutoML 2024_

### Official Review · Reviewer_Y9NU · 2024-03-09

**Potential Impact On The Field Of Automl Rating:** 3
**Technical Quality And Correctness Rating:** 4
**Clarity Rating:** 3

**Summary Of Contributions:**

This paper presents a new approach for automatically choosing a good design for an approach in probabilistic time series forecasting known as "predictive Whittle networks" (PWNs). PWNs are a hybrid structure, consisting of a macro- and a micro-architecture. The proposed approach is capable of optimizing both architectures in conjunction. The results show a benefit of this new approach over hand-crafted baselines.

**Actions Required To Increase Overall Recommendation:**

The authors should detail more clearly how the eventual optimization algorithm operates. Moreover, they should discuss its parameters and what budget this incurs for each of the runs.

**Clarity:**

The paper is mainly well written but misses some important points for me. First, the main approach used for solving the bi-level optimization is not presented clearly. The idea and explanation of what the approach is supposed to do is presented nicely, but the actual approach is missing. Appendix F mentions four lines of pseudo-code that sketch the rough idea. However, I would have liked to see a discussion of the actual approach, especially of its parameter choices. Second, as already mentioned above, the authors do not really discuss how expensive it is to run this approach. Appendix C mentions a total of 5000 GPU hours, but this is not broken down into the cost per dataset. These two points reduce the overall clarity of the paper non-negligibly.

**Overall Review:**

This paper presents a new approach that outperforms hand-crafted approaches. The authors also analyze the performance of their approach with respect to different features and show that despite the overall top performance, there is still room for improvement on an individual level. I appreciate that. The downsides of this paper are that the main algorithmic approach as well as its cost are not very clearly discussed.

**Potential Impact On The Field Of Automl:**

In the context of probabilistic time series prediction, the proposed approach seems to be very valuable, as it allows to search automatically for good architectures. The only downside for me is that the cost for doing so is not properly discussed (or I missed it), making it not entirely clear whether such an approach is cheap and can be used by everyone or not.

**Review Confidence:**

3

**Review Rating:**

8

**Review Summary:**

The approach proposed in this paper seems novel and good to me. It is evaluated fairly and in a diverse manner, making this paper generally very good. However, as some parts are lacking a proper discussion, the paper loses a bit of its value.

## Edit

The author’s updates to the paper convinced me to increase my overall score.

**Technical Quality And Correctness:**

The experiments as well as the conclusions drawn from them appear fair to me. The evaluation encompasses multiple dimensions and thus assesses the performance of the new approach from different angles.

---

### Official Review · Reviewer_WBwa · 2024-03-26

**Potential Impact On The Field Of Automl Rating:** 3
**Technical Quality And Correctness Rating:** 2
**Clarity Rating:** 3

**Summary Of Contributions:**

The paper aims to bring neural architecture search (NAS) to the design of predictive Whittle networks (PWNs) used for probabilistic time-series forecasting, which is currently manually designed. To do so, the authors introduce a bi-level optimization framework: at macro- and micro-level, the authors used DARTS-like procedure to both determine both the operations (in the micro level) and the arrangement of the cells resulting from the micro-level search (in the macro level). Across several tasks, the authors show the superiority of the NAS-designed PWNs compared to the state of the art.

-- Post rebuttal --

I thank the authors for responding to my comments, which partially addressed my concerns. While the new experimental results resolve my concerns somewhat, what remains unknown to me is how much the performance improvement roots from the addiitonal compute. While it's true that bi-level search outperforms one-level search, it also seems to require more compute and I believe the paper will benefit further to discuss on a compute-matched basis for better understanding of the performance-cost tradeoff. However, I do commend authors for their effort and I've increased my rating.

**Actions Required To Increase Overall Recommendation:**

Please address my concerns by providing a more direct ablation study -- I will reconsider a rating upon seeing a satisfactory author response (and the other reviews).

**Clarity:**

The paper is written clearly and illustrations are useful. I do not see major clarity issue with the paper.

**Overall Review:**

Overall, I think this paper presents a sound idea by straightforwardly adapting NAS methods into a specific problem (probabilistic time series forecasting), and the results seem promising and overall, consistent. As such, I think this paper will be of interest to the people in that relevant community and is of interest to AutoML conference as well. On the flip side, I have concerns on the necessity of the design choices, and I encourage the authors to better justify them through more thorough ablation studies, as this is the core contribution of the paper. Also, as mentioned, methodologically, the paper mainly uses existing ideas from NAS, so I do not deem the methods to be very novel overall (although this is not necessarily a weakness).

Since my concerns about the validity of the design choices touch upon the core contribution of this paper, I am not ready to recommend acceptance yet until I see more concrete evidence supporting the authors' arguments.

**Potential Impact On The Field Of Automl:**

Probabilistic time series forecasting is an important area and to my knowledge, the existing applications of NAS in this setup are limited. As such, I think this paper is at least one of the first works bringing these two fields together and I can see consistent improvement over the datasets considered. As such, I think at least members of the probabilistic time series forecasting will find this work interesting.

**Review Confidence:**

3

**Review Rating:**

6

**Review Summary:**

Please see overall review.

**Technical Quality And Correctness:**

The proposed approach can be seen as adapting NAS to a specific task with techniques that bear resemblance to well-tested ideas in NAS such as DARTS, and as such I do not foresee major technical problems with the proposed method. However, a major concern of mine is that more thorough ablation studies should be provided to justify the necessity of the bi-level formulation, which is arguably rather complicated to implement. While the authors separate the experiment sections into sub-sections that discuss the use of both macro and micro level results, it seems that the way the authors evaluate the usefulness of each components is rather indirect: from Table 3 and 4, for example, they mainly look at the preference of operation and use strong preference towards and against some operations as justification of their design choices. However, I think a more direct test is simply to run 1) macro-only, with some handpicked micro architecture, 2) micro-only, with some predefined macro architecture and 3) both micro and macro, as the paper is suggesting -- results like this will probably give more confidence in the readers that both components genuinely contribute to the improved performance.

---

### Official Review · Reviewer_nXn5 · 2024-03-29

**Potential Impact On The Field Of Automl Rating:** 1
**Technical Quality And Correctness Rating:** 4
**Clarity Rating:** 3

**Summary Of Contributions:**

This paper proposes to search for the neural architecture of Predictive Whittle Networks (PWN) [1], that is used for time series forecasting. To do that, the authors parameterize both the macro and micro structure of the PWN architecture and deploy standard gradient-based bi-level optimization methods such as DARTS to search for the optimal architecture. Results show that the found architecture is slightly better than the baselines on a couple of standard time series forecasting datasets.

-- References --

[1] Yu et al. Predictive Whittle Networks for Time Series. In UAI 2022

**Actions Required To Increase Overall Recommendation:**

- *Comparative Analysis*: Conduct a thorough comparative analysis with existing NAS approaches that tackle simultaneously the macro and micro search, such as AGNAS, to highlight the unique advantages and limitations of the proposed method.

- *Performance Evaluation*: Perform extensive performance evaluation on a wider range of time series forecasting datasets, including both standard benchmarks and real-world datasets, to demonstrate the robustness and generalizability of the proposed architecture. Some of these datasets could be the ones in Table 2 of the recent Chronos paper (https://arxiv.org/pdf/2403.07815.pdf).

- *Sensitivity Analysis*: Conduct sensitivity analysis to assess the stability and sensitivity of the proposed architecture to variations in hyperparameters and dataset characteristics. The instability of DARTS like approaches has been studied extensively before and these problems are still prominent in many domains when using these NAS optimizers.

- *Alternative Optimization Methods*: Explore alternative optimization methods for neural architecture search beyond DARTS to ensure the robustness and reliability of the proposed approach.

- *More Ablation Studies*: Even though the authors provide some insights on the results and the architectural components chosen by the NAS algorithm, I would still recommend to investigate more thoroughly the underlying reasons why these choices are prominent.

- *Scalability Analysis*: Assess the scalability of the proposed approach to larger and more complex time series forecasting tasks, considering factors such as computational efficiency and memory requirements.

**Clarity:**

There is no major flaw in terms of clarity. There are some typos here and there, however the method description is straightforward and clear enough.

**Overall Review:**

While the application of Neural Architecture Search (NAS) to the time series forecasting domain presents an intriguing avenue for improving predictive models, the impact of this approach appears limited. The paper introduces Predictive Whittle Networks (PWN) as the subject of study, proposing to explore their neural architecture using gradient-based bi-level optimization methods like DARTS. However, despite the novelty in applying NAS techniques to time series forecasting, the contribution seems somewhat overshadowed by existing approaches. For instance, prior works such as AGNAS have already delved into the exploration of both macro and micro structures in neural architectures.

Furthermore, while the paper reports marginal improvements over baseline models on standard time series forecasting datasets, the significance of these gains remains questionable. The improvements as shown in Table 2 are statistically not significant, which raises concerns about the practical relevance and generalizability of the proposed architecture. Without substantial performance enhancements, the broader impact of this research on real-world time series forecasting applications may be limited. Moreover, it is not clear to me from the paper what is the computational cost of the NAS search and final training of the found model. It would be great if the authors emphasize this better.

Moreover, the choice of DARTS as the optimization method for searching the optimal architecture raises concerns about the robustness and scalability of the proposed approach. While DARTS is a popular choice for architecture search due to its efficiency, it often suffers from issues such as instability and sensitivity to hyperparameters. Without thorough experimentation and comparison with alternative optimization techniques, the reliability and effectiveness of the proposed method remain uncertain.

**Potential Impact On The Field Of Automl:**

Even though the application of NAS to the time series domain might be useful to boost performance, I do not see much potential impact coming from this approach. Algorithmically, there are previous works that try to search for both macro and micro structures, such as AGNAS [1], therefore I do not see much contribution in that regard.

-- References --

[1] Sun et al. AGNAS: Attention-Guided Micro- and Macro-Architecture Search. In ICML 2022

**Review Confidence:**

4

**Review Rating:**

3

**Review Summary:**

Overall, while the paper presents an interesting exploration into the neural architecture of Predictive Whittle Networks for time series forecasting, its contribution appears somewhat incremental and overshadowed by existing approaches. To make a more substantial impact, future research efforts could focus on addressing the limitations of the proposed approach and exploring alternative methods for architecture search in the time series domain.

-- Updated score --

There seem to be technical flaws in the experimental evaluation that need to be addressed before accepting this paper.

**Technical Quality And Correctness:**

I do not see any flaws in the experimental evaluations conducted throughout the paper. Nevertheless, in most of the cases in the results of Table 2, the differences compared to baselines are not statistically significant.

---

### Official Review · Reviewer_Aa87 · 2024-03-30

**Potential Impact On The Field Of Automl Rating:** 3
**Technical Quality And Correctness Rating:** 3
**Clarity Rating:** 4
**Actions Required To Increase Overall Recommendation:** 1. More discussion or experiments on …

**Summary Of Contributions:**

The paper presents a bi-level optimization to automatically tune predictive Whittle networks, where the first level is the macro architecture of the nodes in the graph, and the second level is the actual weights (micro architecture) between each connected node. Experiment results demonstrate the effectiveness of the proposed approach.

**Clarity:**

The paper is generally well-written and most of the details are nicely explained, as well as the key design rationals.

**Overall Review:**

Overall, despite some limited evaluation of the overhead of the approach, I feel that the work is technically sound and the result promising. The only minor point is to provide more justification about the additional overhead and whether any other mechanisms have been applied to tackle such.

**Potential Impact On The Field Of Automl:**

Bi-level optimization has been an increasingly popular direction for AutoML, and hence this work poses a good impact on the community.

**Review Confidence:**

4

**Review Rating:**

8

**Review Summary:**

I believe that this paper can have a good impact on the community and promote the bi-level optimization of AutoML. The issue in the evaluation is mostly fixable in the camera-ready version.

**Technical Quality And Correctness:**

The proposed approach is technically sound and of good quality. The idea of bi-level optimization is natural given the hierarchical relation of the macro/micro architecture for the predictive Whittle networks. However, one key concern is that the bi-level optimization can be rather expensive, considering that the evaluation of AutoML is already quite time-consuming. The only method that the paper has relied on to that issue is by using one-step gradient descent. This raises a question: Is the benefit provided by the bi-level optimization really more significant than the additional overhead it introduces? Therefore, it would be good to examine the optimization overhead of the approach.

---

### Meta-Review · Area_Chair_ZfEp · 2024-04-22

**Paper Recommendation:** Accept to workshop track
**Confidence:** 4

**Metareview:**

The paper proposes an architecture search approach for probabilistic time-series forecasting. In particular, the method uses an approach similar to DARTS to search for the micro and macro levels operations.

The reviews are mixed, generally the reviewers saw some quality of the paper in the novel application of Darts to the probabilistic time series forecasting use-case. However, some reviewers also complained about the novelty of the method, issues with empirical results and potential issue with the computational cost.

While most issues related to the computational cost has been addressed by the authors, issues remains on the novelty of the method and its empirical assessment. In particular, the paper seems to be lacking in this aspect as the only baseline are PWN variants but no deep-learning, statistical or AutoML methods are considered making the comparison a bit odd. When looking at the PWN paper, other baselines are compared including deep-learning methods but the results seems inconsistent with the current paper as the accuracy numbers of PWN baselines widely differ.

One reviewer also noted that the results may not be statistically significant, the authors answered that their methods outperforms all other baseline when taking the the mean of their methods against the mean of a baseline minus the std of the baseline. However, this argument is dubious given that their method exhibits much larger variance (often twice larger), in fact if one takes the mean of their method plus the std of their method, then the number of wins is quite small.

For those reasons, the paper seems to require strenthening in the empirical results, I would encourage the authors to consider Autogluon timeseries [2] and ensemble of statistical forecast methods [3] and take care to analysis the statistical strength of their results, possibly using critical diagrams [4] to make statistical comparisons of multiple methods over multiple datasets.

[1] Predictive Whittle Networks for Time Series. Yu et al.
[2] https://auto.gluon.ai/stable/tutorials/timeseries/index.html
[3] https://github.com/Nixtla/statsforecast
[4] https://www.jmlr.org/papers/volume7/demsar06a/demsar06a.pdf

---

### Decision · Program_Chairs · 2024-04-29

**Decision:**

Accept

**Comment:**

Thank you for submitting your paper. We had a very thorough discussion about this paper, including the AC, PC chairs, and the GC. There were several concerns regarding the validity of the results, incl. missing comparisons against more baselines (also statistical models) and comparisons against other DL models but with more epochs (15k as in the original PWN paper); also see AC comments. We checked the paper's results against other sources (incl. further papers) -- for example, the results in this paper look promising in view of the results Monash Time Series Forecasting Repository. Nevertheless, we would like to ask the authors to run more experiments (e.g., with 15k epochs and more baselines), check whether all claims still hold true, and add them to the CRC of the paper (i.e., at the very least, to the appendix).

Overall, we are happy to tell you that we accept your paper to the main track. See you in Paris.